# The Effect of Stabling Routines on Potential Behavioural Indicators of Affective State in Horses and Their Use in Assessing Quality of Life

**DOI:** 10.3390/ani13061065

**Published:** 2023-03-15

**Authors:** Ella Bradshaw-Wiley, Hayley Randle

**Affiliations:** School of Agricultural, Environmental and Veterinary Sciences, Charles Sturt University, Wagga Wagga, NSW 2650, Australia

**Keywords:** equine, behaviour, welfare, affective state, quality of life, stabling

## Abstract

**Simple Summary:**

The keeping and use of horses have become of increased interest to the public due to welfare concerns. It is therefore vitally important to better understand the impacts on the horse’s emotional state and how to measure and use observed behaviours to determine the effects of common husbandry practices on the horse. This will enable steps to be taken to improve equine quality of life and ensure the social licence to operate within the horse industry. In order to achieve this, reliable animal-based behavioural indicators of welfare that can be used in industry are needed. The behaviour of horses kept on day-stabling routines was compared to that of horses kept on night-stabling routines. Eight behaviours including ear movement and locomotory leg movements as well as yawning, recumbency, and non-nutritive chewing occurred significantly more often in horses on a night-stabling routine. These behaviours have been identified as potential indicators of affective state (the animal’s underlying emotional state) and equine welfare that can be used in dynamic quality of life assessments.

**Abstract:**

Increasing interest in equine welfare has emphasised the need for objective and reliable behavioural indicators of horses’ affective state. However, research has yielded mixed results regarding behaviours suited for industry use largely because they are subject to anthropomorphic interpretation. Stabling is commonly used to manage domesticated horses despite research indicating that it can negatively impact horse welfare, but its effect on their affective state is yet to be quantified. Ten adult horses (11.8 ± 4.4 years) were observed either on a day- (DS) or night-stabling (NS) schedule over two consecutive 24 h periods. NS horses were kept confined for significantly longer (13.60 ± 0.04 h) than DS horses (7.73 ± 0.07; *t*_7_ = 5.70; *p* = 0.0004). Eight behaviours occurred significantly more often during NS than DS: forward ears (*t*_7_ = 3.32; *p* = 0.001), neutral ears (*t*_7_ = 3.47; *p* = 0.001), stepping forward (*t*_7_ = 2.62; *p* = 0.001), stepping laterally (*t*_7_ = 2.39; *p* = 0.001), sternal recumbency (*t*_7_ = 2.64; *p* = 0.001), yawning (*t*_7_ = 2.69; *p* = 0.001), non-nutritive chewing (*t*_7_ = 2.49; *p* = 0.001), and closing eyes (*t*_7_ = 2.71; *p* = 0.001). These behaviours may be candidates for indicators that can be used to determine the affective state in horses and subsequently be used to assess equine quality of life and to optimise individual horse welfare.

## 1. Introduction

The welfare of horses has been subject to increasing scrutiny by the general public due to high-profile sporting events where horse management and use have at times appeared to be substantially less than optimal. Failure to appropriately manage horse welfare leaves the industry with a tenuous social licence to operate (SLO). It is therefore increasingly important to review contemporary and emerging management practices and their alignment with the promotion and safeguarding of good welfare [1]. Whilst much of the equine industry and the scientific community recognise the need for a reliable and robust method of welfare assessment [2,3,4], the complexity of this task is evident from the still-evolving nature of applicable welfare assessment tools that will enable reliable assessment of the horse’s physiological and psychological needs [5,6]. Despite a growing appreciation of animal-based indicators of welfare, difficulties remain with accurately determining objective measures of an animal’s emotional state in general, and of positive affective state in particular [1]. In the absence of validated measures for accurately identifying equine affective state, the most common method of evaluating equine welfare is behavioural observation, therefore making the identification and validation of behaviours reliably linked to mental state critically important [7].

Although keeping horses in the paddock full-time is often considered preferable [8], this may not be possible due to restrictions on resources such as space and feed, resulting in many horses being stabled for at least part of the day. Some horse keepers consider stabling a suitable alternative, or in some cases, the ideal method of housing for horses [9]. Common management considerations of owners include available land, space, cost, access to riding facilities, and proximity to events or veterinary clinics. This may be compounded by a lack of knowledge regarding practices that optimise horse welfare, leading to impacts on equine quality of life. Meanwhile, constraints may also be placed upon horses to manage their dietary or physical requirements, e.g., yarding (small, confined outdoor pen) [10,11].

Stabling has been linked to significant impacts on the horse’s physiological health, as evidenced by the increase in clinical conditions such as equine asthma [12,13,14] and equine gastric ulceration syndrome (EGUS) [15]. However, individuals and businesses offering housing for horses may not find it easy to provide more appropriate housing or understand the need to do so [9]. Many facilities subdivide the land available into small areas to increase the number of horses that can be provided for [16], and as Yarnell et al. [17] observed, this can lead to a reduction in group/herd size and may even favour keeping horses singly, which has also been reported to be contrary to good welfare. Therefore, it is important that modern horse management practices are routinely scrutinised and investigated for opportunities to optimise the horse’s home environment. Whilst there has been research conducted regarding the behaviour of stabled horses [18,19,20,21], particular emphasis has been put on stereotypies [22,23,24] or specific behaviours [25], with little regarding the effect of different stabling routines on equine behaviour and welfare. In addition to this, the effects of light and circadian rhythm on horse behaviour lack sufficient research to be quantifiable.

When using evidence-based research outcomes, the optimal form of horse housing is commonly regarded as large paddocks that enable locomotory behaviour and provide access to conspecifics, with adequate shelter and sufficient nutritional content in the grass to minimise or negate the requirement for supplementary feeding [9,19]. When comparing the time budgets of horses kept in modern housing situations, a dramatic contrast to their feral counterparts is shown, with significantly reduced locomotion, grazing behaviours, and conspecific interactions [18,26]. The Five Domains was proposed in 1994 as a comprehensive method of welfare assessment due to its ability to provide a systematic and structured approach to animal welfare assessment and management with an increased focus on the impact of animal affective state [6]. The adoption of this welfare framework has allowed researchers to further consider an individual’s mental state [5], and current evidence exists that stabling can have a detrimental effect on the mental welfare of the horse [21]. These combined effects on the physiological and psychological wellbeing of the horse can significantly impact their overall welfare and quality of life [17] and highlight the need to investigate the effects of stabling on equine welfare and, consequently, quality of life.

There is currently no standardised equine quality of life (EQoL) assessment framework in place, and importance has been put upon the development of a system that uses evidence-based and observable behaviours that can reliably reflect the individuals’ affective state [27]. This will allow for the preferences of the individual to be better recognised and catered for, meaning that common husbandry procedures, such as stabling, can be optimised to better suit the individual, focusing on a ‘best fit’ model rather than a rigid focus on a ‘best practice’ approach [28].

The aim of this study was to determine the effect of stabling routines on equine behaviour in order to identify potential indicators of their affective state that may be suitable measures of equine quality of life with further focused research and validation. Two common stabling routines where turnout is limited were compared: horses stabled during the day and turned out overnight and horses turned out during the day and stabled overnight. The hypothesis that stabling routines would not have a significant effect on horse behaviour was investigated.

Behaviours were observed via the use of continuously recording video cameras placed discretely in the stable and classified according to an ethogram developed during an earlier pilot study of five horses. The significant behaviours identified by this study were then investigated in the literature for potential links to affective states in horses as well as other species.

## 2. Materials and Methods

### 2.1. Subjects

The horses used in this study were recruited from the existing pool of student horses boarded at the Charles Sturt University Equine Centre. Participation in the study was on a voluntary basis from the owners, for which written consent was obtained. Ten geldings with a mean (standard deviation) age of 11.8 (4.4) and various breeds were used for this study (Table 1). All horses were considered to be pleasure and/or low-level performance horses in light to heavy dressage and jumping work as described by the National Research Council (NRC) 2007 guidelines [29]. The mean (SD) body condition score (BCS) was 3.35 (0.45) at the commencement of the study. The horses included in the study were not suffering from any illness, disease, injury, or lameness as defined by the owners.

### 2.2. Horse Housing and Management

Two stabling routines were investigated in this study. Five horses were stabled overnight from approximately 17:00 to 09:00 and turned out into a two-acre paddock shared with 3–5 conspecifics for the remainder of the day. The remaining five horses were stabled during daylight hours from approximately 09:00 to 17:00 and turned out into the same two-acre shared paddocks with three to four conspecifics overnight. The stabling facility used was located at the Charles Sturt University Equine Boarding Centre. Stables were 4 m × 4 m and bedded approximately 25 cm deep with sawdust. Horses were able to make olfactory, visual, and restricted tactile contact with neighbouring horses on two to three sides of the stable. All horse management decisions including stabling routine and timings, nutrition, and workload were determined by the owner, and no changes were made to the normal routine as a result of participation in this study. All horses were stabled at the CSU Equine Centre prior to their involvement in the study and were observed in their familiar home environment/stable. Stabling allocations, paddock allocations, and routines were predetermined by the CSU Equine Centre staff, and no changes were made to this when participating in the study. Throughout this study, all horses had at least one horse in a neighbouring stable whilst being observed.

When stabled, forage was provided in the form of hay via hay nets or on the floor of the stable according to owner preference, and fresh water was provided in one or more buckets. Boarding facility regulations necessitated that horses be turned out for half the day, and when turned out in the paddock, horses had access to three or four conspecifics and ad libitum access to water but limited pasture. The available pasture was known to be insufficient in nutritional content, quantity, and quality to adequately meet the forage requirements of an individual horse, and owners were not permitted by the facility to provide supplementary feeding during turnout to avoid incidental feeding of other horses and the potential risk of increased injury.

Due to the temperate climate of the area, the stabling facility had an open and back-to-back layout, meaning that stables were not enclosed to the outer environment and had an open view of the surrounding facility. Therefore, during the day, natural lighting was the main source of light. At night-time, there were lights available to handlers for use when at the facility, which were extinguished from 10:00 p.m. Sunrise was at approximately 7:00 a.m. and sunset at approximately 5:00 p.m. during the time of this study.

### 2.3. Materials

A 4-channel Techview 720p AHD DVR recorder with four indoor/outdoor 720p cameras equipped with infrared LEDs to allow for night-time recording was used. Data from the Techview DVR were transferred onto a 2 TB Seagate external hard drive to allow for the viewing of footage on a Dell Latitude 5400 computer via the VLC Media Player application.

### 2.4. Procedure

As this study was a purely observational study, there was no intervention by researchers, and horses were observed in their familiar stabled environments. All horses were filmed over a 48 h period, allowing data capture of two successive occurrences of stabling per horse.

A preliminary investigation was conducted on five horses to determine the number of cameras needed per animal in order to effectively capture behavioural data and to aid in the development of the ethogram for this study. The data generated in the preliminary investigation was not included in the final study due to changes made to the experimental procedure and design.

Two cameras were used per stable, and cameras were set up at the highest point of the stable walls in diametrically opposing corners to minimise interference with or by the horse as well as allow for the maximisation of space captured. This allowed for two horses to be recorded per 48 h period. Researchers contacted the owners prior to the setup and installation of the cameras to coordinate a time to do so during which the horse was not being stabled as well as to notify them when the recording would begin. Recording began from the moment camera installation was completed and was kept on for the 48 h period in which the horses were being observed.

Once camera installation had been completed, horse owners were notified that the recording period had begun and advised to continue with their normal stabling and husbandry routines. Recording ceased after 48 h at the conclusion of the observation period, and horses were removed from the stable. Recordings were dated and timestamped by the Techview DVR recorder to allow for identification when removed and backed up onto an external hard drive.

### 2.5. Data Extraction and Collation

Recordings were partitioned by the Techiew DVR system (Jaycar Electronics, Sydney, Australia) into 45 min blocks of footage. These video data files were downloaded from the DVR system onto the external hard drive where the data files were then organised by horse, date, and time once uploaded to the computer.

A continuous time sampling method was used to assess the behaviour exhibited by all horses during the stabling period and viewed using VLC Media Player. The samples were analysed in ten-minute blocks, commencing with the first ten minutes of the horse in the box and post the owner departing the stable upon completion of normal husbandry practices and ending with the last ten minutes before either the horse was removed from the box or the stable was entered by the handler. For each hour between entry and departure from the stables, an interim block was chosen for scrutiny using the Google online random number generator. The total number of blocks analysed per horse was dependent on the length of time spent in the stable, as dictated by owner management preferences.

Each sample video was observed continuously, and the subjects’ behaviours were recorded instantaneously and classified according to the ethogram comprising 52 possible behaviours developed during the pilot study of 5 horses, with behaviours categorised as per Kiley-Worthington (1997) [18] (Table 2). Individuals were also observed for their response to humans entering their stable for voluntary approach. Frequencies of behavioural occurrences were derived for each sample video and recorded into an MS EXCEL v16.0 (Microsoft, Redmond, WA, USA) spreadsheet for collation.

### 2.6. Data Analysis

Behaviours were categorised for time budget analysis (Table 2) for comparison with existing published equine time budgets [30]. This was performed with the aim of identifying potential differences between the two stabling routines as well as potential differences between the horses used for this research and full-time stabled horses observed by Kiley-Worthington in 1989 [30]. The 52 observed behaviours were categorised into one of seven possible groups (eat, stand, lie, locomotory, non-locomotory, elimination, or other); however, this analysis ultimately excluded eight of the behaviours that were categorised as “Unspecified”. Seven of the behaviours were excluded as they were able to be performed within any behavioural classification, for example, ears forward whilst head up, whilst the eighth represented no visual contact with the horse and was therefore unable to be considered a performed behaviour.

### 2.7. Statistical Analysis

The data analysis involved tabulating the results in MS Excel v16.0, followed by statistical analysis in R version 3.6.2 [31]. The normality of the data was verified through a Shapiro–Wilk normality test. To compare the behaviours exhibited by day- and night-stabled horses, a series of unpaired *t*-tests were used. The equality of variance was assessed using an F-test, and a Holm–Bonferroni correction was implemented to minimise the risk of Type I errors. A significance level of *p* < 0.05 was consistently applied. Moreover, a hierarchical clustering analysis method was employed using the hclust function of R to identify potential clusters of behaviours observed in the two test conditions, i.e., stabled at night and stabled during the day.

### 2.8. Ethical Approval

The research undertaken was approved by the Charles Sturt University (CSU) Animal Care and Ethics Committee, authority no. A18043.

## 3. Results

Observations of 52 different behaviours across nine horses over 12 days and 576 h of footage resulted in 13,778 behavioural occurrences recorded. The mean (SD) number of behaviours performed was 14.8 (1.1) per time sample blocks analysed per horse in the night-stabled routine and 8.5 (1.7) per time sample blocks analysed for horses in the day-stabled routine. This yielded a mean (SD) of 1531 (688.8) observations per horse. Horse 10 was excluded from the study due to owner mismanagement that resulted in stable confinement for the entire 48 h period, a decision that was made independently of the study and meant that observations would have not been consistent with other horses receiving turnout.

### 3.1. Time Budgets

The 52 observed behaviours were organised into seven categories (Table 2). Behaviours listed as belonging to the unspecified category were excluded from the time budget analysis as these could be performed in conjunction with other behaviours. A bar chart was constructed to illustrate the time budgets of stabling routines and the total population over their respective 48 h sampling period (Figure 1).

### 3.2. Significant Behaviours

Horses kept in the stables overnight were housed for a significantly longer period (13.60 ± 0.04 h) than horses kept in stables during the day (7.73 ± 0.07 h; *t_7_* = 5.70; *p* = 0.0004) and displayed significantly more total behaviours on average per hour of stabling (*t_7_* = 2.70; *p* = 0.03).

A series of unpaired *t*-tests were also used to compare the frequencies of behaviours occurring with stabling routines, and 11 behaviours were found to have a significant difference. This was reduced to eight after the Holm–Bonferroni correction was applied. No horses demonstrated stereotypical behaviours such as windsucking, weaving, or box walking, whilst all horses demonstrated a positive approach when humans entered the stable. Significant increases in the frequency of ear movements (forward ears, *t_7_* = 3.32, *p* = 0.001; neutral ears, *t_7_* = 3.47, *p* = 0.001; locomotory behaviours (stepping forward, *t_7_* = 2.62, *p* = 0.001; stepping laterally, *t_7_* = 2.40, *p* = 0.001); and behaviours classified as ‘Other’ (yawning, *t_7_* = 2.70, *p* = 0.001; non-nutritive chewing, *t_7_* = 2.49, *p* = 0.001) were seen in night-stabled horses. Other behaviours that showed significant increases for night-stabled horses were closing of the eyes (*t_7_* = 2.69; *p* = 0.001) and sternal recumbency (facing sideways, t_7_ = 2.640; *p* = 0.001). Significant behaviours and the number of occurrences were compared using a bar chart (Figure 2).

Hierarchical cluster analysis was conducted to identify potential behavioural clusters that occur under the different stabling conditions and may help indicate relative affective states of behaviours of significance with better researched and understood behaviours. Figure 3 and Figure 4 visually illustrate clusters of behaviours for the night-stabling and day-stabling routines, respectively, generated by hierarchal clustering analysis.

## 4. Discussion

For a behavioural observation to be useful in welfare assessments, it needs to be easily observed and validated [1], which has meant that high-intensity behaviours such as overt aggression or stereotypies such as box-walking and weaving are easily classified and associated with compromised welfare, suggesting negative affective states [22]. However, given that welfare is experienced differently by individuals, the absence of obvious negative affective states does not indicate that the individual is experiencing ‘good’ welfare, nor does the absence of obvious positive affective states suggest poor welfare [32]. Unsurprisingly, much difficulty has been encountered in determining the possible affective state associated with more subtle equine behaviours [33]. Therefore, behavioural observations were used in acknowledgment that other physiological measures such as heart rate (HR), heart rate variability (HRV), and salivary cortisol are not accessible to the average horse owner and still require sufficient validation when used to measure behaviour, making it important to see if observations and subsequent welfare assessments could be used independently.

Whilst the video capacity did not permit an accurate assessment of eye blink rate in this study, instances of eyes closed were able to be captured and analysed. If the spontaneous blink rate (SBR) is required in future studies, the study design would require adjustment to accommodate the limitations experienced in this instance. However, horses stabled at night displayed significantly more instances of eyes closed than those stabled during the day. This appears to coincide with the significantly greater number of night-stabled horses that would sleep in a sternal recumbent position. When examining the dendrogram for behavioural clustering of blinking behaviours, day-stabled horses did show some correlation between eyes closed and stretching behaviours. However, when examining the dendrogram of night-stabled horses, there was a stronger correlation between the eyes closed behaviour and the behaviour of facing away from the stable door with their heads lowered (negative internal). A likely explanation for this is that the horse is better able to achieve a restful state when external stimulation is at a minimum, allowing the horse to feel comfortable enough to face into the box.

Unlike humans, horses have adapted to minimal sleep patterns and only sleep for approximately four hours a day. Whilst horses have evolved to sleep standing up via the use of the stay apparatus, effective rapid eye movement (REM) sleep only occurs during sternal or lateral recumbency and is an important requirement for overall horse health, both physiologically and psychologically [34]. Multiple factors affect recumbency in horses including bedding substrate [34,35] and stable size [36] and may also be influenced by equine stress levels [34]. Seven of the nine horses were observed in sternal or lateral recumbency during the time periods analysed. Horses stabled overnight were significantly more likely to display sternal recumbency, which may indicate a positive affective state given the option to perform this behaviour whilst in shared turnout with conspecifics, but were relaxed enough with their surrounding environment to pursue a full rest state when stabled. This suggestion is supported by the findings of Mazzola et al. in 2021 [37], which found horses that were brought in overnight for stabling had better sleep quality and reduced cortisol levels compared to horses kept naturally. When considering sternal or lateral recumbency as a potential cluster or paired behaviour, the highest level of correlation in the dendrogram for night-stabled horses is seen between standing negative facing a companion and sternal recumbency facing a companion. It is plausible that this is a vital component of the individual ethological need for companionship in order to achieve an effective resting state. When viewing the dendrogram for day-stabled horses, the strongest correlation was between a form of sternal recumbency (performed facing out of the stable) and head nodding; however, it is not yet clear how or why these behaviours may be related.

The behavioural analyses revealed that the stabling regime impacts horse activity. Horses stabled at night spent significantly longer in the stable than when turned out in the paddock with herd mates than horses stabled during the day. It could be suggested, therefore, that if owners are looking for a way to maximise a horse’s access to turnout, it may be beneficial to stable horses during the day and turn them out. This may be of further benefit to horses housed in hot or humid conditions as it allows the horse to be turned out during the coolest part of the day. This may also support the use of other beneficial amenities such as misting equipment and fans in warmer climates, pending availability and stable design, without completely restricting the horse from turnout.

Horses kept in the stable overnight were stabled for an average of 13.5 h, whilst horses kept stabled during the day were only stabled for an average of 7.75 h. Whilst time budgets of night- versus day-stabled horses did not show substantial differences in the types of behaviours or stereotypical behaviour, horses stabled overnight displayed more leg movement, such as stepping forward and laterally, which may indicate a milder version of barrier frustration behaviours such as box walking and weaving [38] and therefore indicate links to a negative affective state. When considering the dendrogram for night-stabled horses shown in Figure 4, the closest relationships were between stepping forward and increased tail swishing and between lateral stepping and increased lateral head movements. This may suggest increased movement and activity from the horse overall, which is perhaps indicative of increasing restlessness or frustration due to confinement. In contrast to this, Figure 3 for day-stabled horses showed that forward and lateral steps had closer relationships to rested hind legs and foraging, respectively, behaviours more commonly associated with rest and relaxation [39]. However, when comparing the time budget of the sample horses in this study to traditional time budgets of full-time stabled horses published by Kiley-Worthington in 1989 [30], it is interesting to note that the proportion of time spent eating is far greater for the sample horses. This may be due to changes in management practices since 1989 that have allowed horse owners to be more aware of the need for adequate access to roughage sources during confinement. Eight of the nine sample horses were provided with roughage during stabling, and half of those horses received their roughage in a slow-feeding hay net, which increased the average time spent eating.

Ear positioning is often discussed when observing horse behaviour as a relative indicator of mood or affective state [40,41]; however, the frequency of ear movement displayed by horses in their resting environment has received limited investigation. When investigating the literature on other species, ear position and frequency of movement have been highlighted as being indicative of a negative affective state in previous studies with sheep [42]. Given the proposed relationship between ear position and assumed emotional states such as backward pinned ears being associated with discomfort, pain, or stress in horses [40] and sheep [42], rapid shifting between positions may provide further insight into the horse’s affective state. Increased ear activity was noted in horses stabled overnight, with ears moving between the forward and neutral positions with significantly increased frequency compared to those stabled during the day. Previous research into the positioning of the ears focused primarily on placement when performing behaviours under saddle [41,43,44], experiencing fear or pain [3,41], or as an indicator of the horse’s attention [45]. However, studies have linked the decreased frequency of ear movement to positive attitudes towards humans [45], and therefore, it should not be discounted in the identification of positive affective states. It is important to consider if the increased frequency of ear movement for night-stabled horses may be potentially explained by the decreased visibility conditions increasing the horse’s sensitivity to auditory stimuli. This proposal aligns with Hartman and Greenings’ demonstration in 2019 [25] of the positive influence of auditory stimulation on behaviours such as recumbency in stabled horses.

Non-nutritive chewing behaviour has been previously highlighted in horses as being a response to stress [46], contravening industry associations that this behaviour is indicative of relaxation, learning, cognitive processing, or submission in the horse. However, research conducted by Lie and Newberry (2018) [46] showed that non-nutritive chewing behaviour was seen following aggressive conspecific interactions by both parties, with particularly higher prevalence when observing the aggressor, refuting the suggestion that the behaviour is based on submission. Instead, it is believed that non-nutritive chewing occurs when parasympathetic activity increases following a stressful event. Therefore, the increased prevalence of non-nutritive chewing behaviours observed in night-stabled horses may be indicative of increased stress levels and, subsequently, a negative affective state. When considering the dendrogram for day-stabled horses in Figure 3, non-nutritive chewing was most closely related to drinking, lending further support to the idea that non-nutritive chewing occurs with reactivation of the parasympathetic nervous system, which prioritises digestive processes over the fight/flight response [47]. Similarly, Figure 4 for night-stabled horses shows non-nutritive chewing as being most closely related to head-lowering behaviour, further linking it to an activity performed after experiencing a stressful state. Day-stabled horses also demonstrated a moderate link to yawning, which also has links to negative affective states.

Incidences of yawning were similarly seen to increase in frequency for night-stabled horses; however, its role in the identification of affective state remains complex due to the determinants of yawning behaviour remaining unknown [48]. Whilst yawning behaviours have been shown to have a relationship to sleep in multiple species [49,50,51], similar research exists to show an increase in the prevalence of yawning behaviours following stressful and emotional interactions as well [51,52,53]. Fureix et al. (2011) [48] investigated the relationship between yawning behaviour and stereotypic behaviours in an effort to further quantify the significance of yawning in horses. The results indicated there was a positive correlation between yawning frequency and the performance of stereotypical behaviours, suggesting potential links to poor welfare or a negative affective state. This is further supported by Gorecka-Bruzda et al.’s (2016) [53] findings, in which yawning behaviours were investigated in both domesticated horses and Przewalski horses for potential causation, with findings demonstrating a positive relationship between the frequency of yawning behaviour and stressful social situations.

Whilst stabling has been linked to decreased welfare states in horses due to the inability to engage in natural grazing behaviour and interactions with conspecifics, the horses used in this study did not display any stereotypies commonly associated with stabling, such as weaving, box walking, or crib biting. This may suggest that the provision of even limited turnout access could improve the negative underlying factors that are believed to initiate stereotypical behaviour in some horses, therefore preventing them from originating and becoming consolidated reactions to suboptimal conditions [54,55]. Further to this, all horses used in this study displayed voluntary approach behaviour to the handlers retrieving them from the stable. The voluntary approach response in horses has previously been used as an indicator of a positive affective state in horses and is indicative of their previous human–horse interactions [56]. Whilst a voluntary approach to humans can be considered ideal and a potential indicator of the horse’s affective state, more research needs to be carried out to verify whether this response is due to a positive affective state related to their housing and management, due to their positive association of the presence of a human with their removal from the stable, or simply a conditioned response to human approach regardless of affective state or environmental conditions and management.

The behaviours identified in this study may assist with the identification of affective states in horses with further research and validation to ensure behavioural indicators are robust and reliable across contexts. With increased knowledge of behavioural indicators of affective state, horse owners and managers may be able to better identify individual perceptions of common husbandry practices and make adjustments to better suit the needs of the individual, therefore improving the day-to-day quality of life of the animal.

Some limitations to this study were experienced due to the variability in owner management practices, which may have had an overall effect on behaviour. One horse was removed from the study due to the owner changing their management routine to that of a full-time stabled horse partway through the study, thereby necessitating data generated by that horse for exclusion. The variability between horses in the type of feed/forage provided, how the feed was delivered, exercise routines, and total time spent stabled could be controlled in future studies, resulting in a more homogenous sample population. The small sample size of the population was also considered to be a limitation of the study design.

## 5. Conclusions

Given the fact that stabling is a commonly practiced, accepted, and even promoted or required component of modern horse management, efforts should be made to optimise approaches to housing horses in order to alleviate the stress experienced by the horse rather than focusing on the inherently negative aspects associated with stabling. Whilst it is not yet possible to reliably categorise all behaviours as indicators of positive or negative affective states, specific behaviours such as non-locomotory leg movement as well as ear position and frequency of movement are already being identified for future use in dynamic contemporary welfare assessments that are applicable to all horses and across multiple contexts in which they are displayed.

## Figures and Tables

**Figure 1 animals-13-01065-f001:**
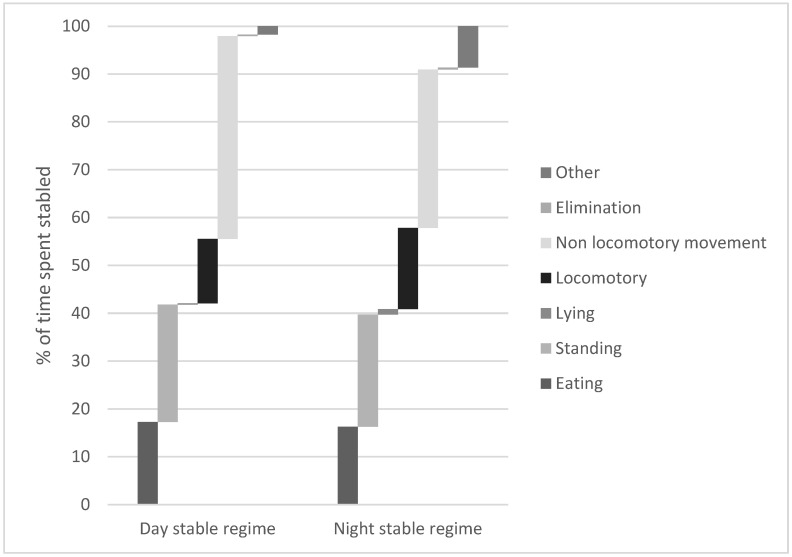
Total behaviours expressed for behaviour categories outlined in Table 2 as a percentage of total behaviours displayed during stabling over the recorded time period.

**Figure 2 animals-13-01065-f002:**
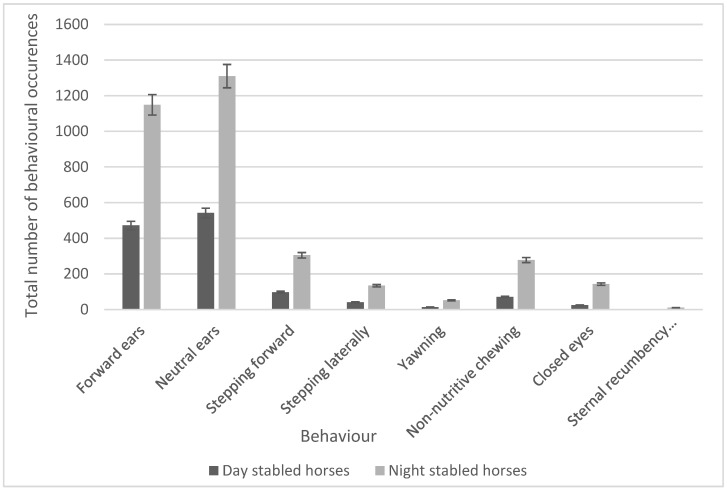
Comparison of the significant behaviours between day- vs. night-stabled horses.

**Figure 3 animals-13-01065-f003:**
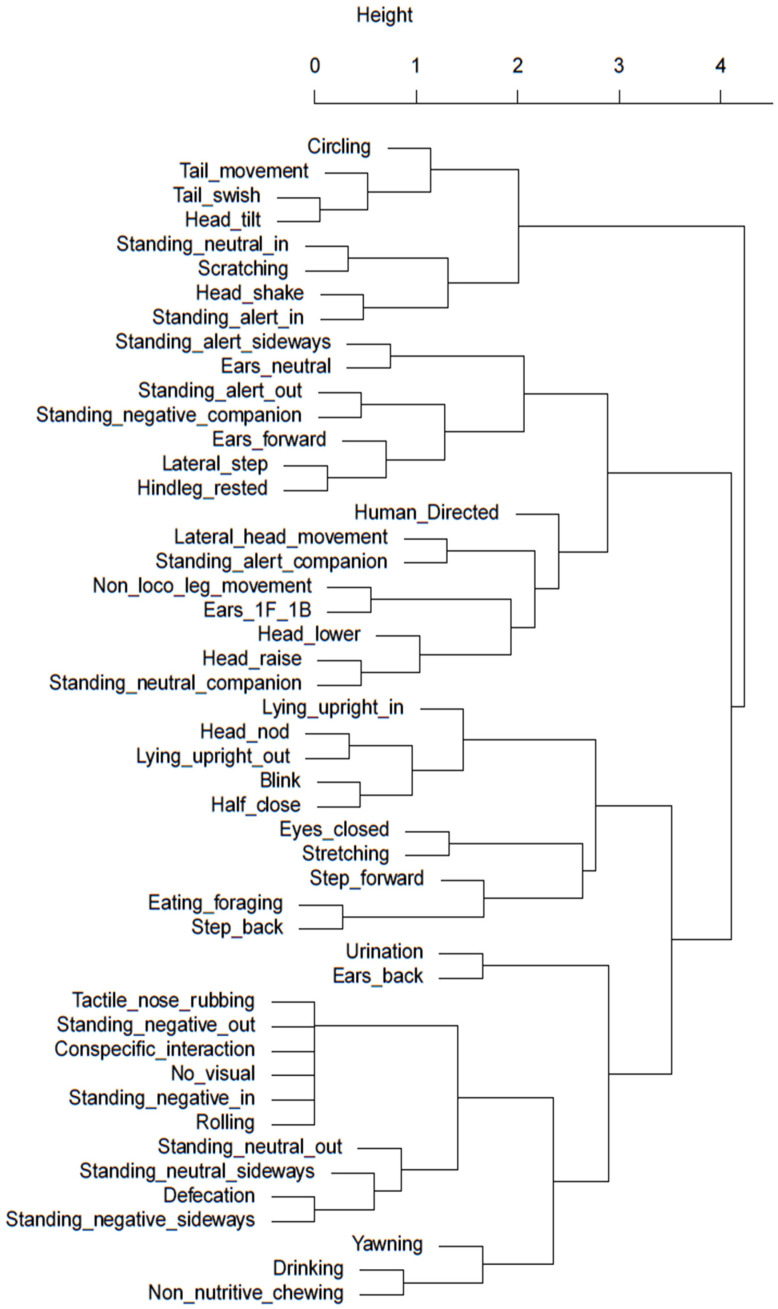
Cluster dendrogram of day-stabled horses. Behaviours with smaller heights in shared clades demonstrate the highest level of correlation. A greater distance between clades and increasing height of the *y*-axis indicate a weaker relationship.

**Figure 4 animals-13-01065-f004:**
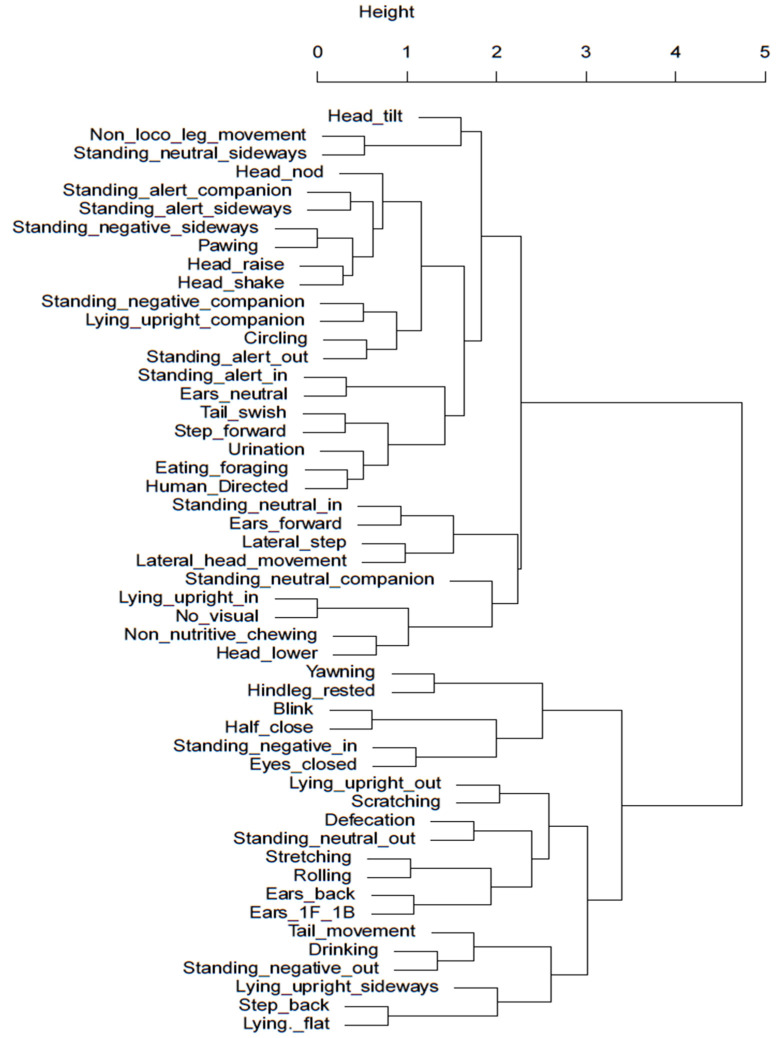
Cluster dendrogram of night-stabled horses. Behaviours with smaller heights in shared clades demonstrate the highest level of correlation. A greater distance between clades and increasing height of the *y*-axis indicate a weaker relationship.

**Table 1 animals-13-01065-t001:** Horse details.

Horse	Age (Years)	Breed	Sex	BCS	Workload	Stabling Routine
**Horse 1**	8	WB	G	3.75	Moderate	Night
**Horse 2**	14	WBX	G	3	Heavy	Night
**Horse 3**	6	WBX	G	3.5	Light	Night
**Horse 4**	16	ASH	G	3	Moderate	Day
**Horse 5**	9	QHX	G	4	Moderate	Day
**Horse 6**	12	WBX	G	3.5	Moderate	Day
**Horse 7**	9	PREX	G	3.5	Heavy	Night
**Horse 8**	21	ASHX	G	2.5	Light	Night
**Horse 9**	8	TB	G	4	Light	Day
**Horse 10**	15	QHX	G	3	Moderate	Day

Breed: WB, Warmblood; WBX, Warmblood cross; ASH, Australian Stock Horse; ASHX, Australian Stock Horse cross; QHX, Quarter Horse cross; PREX, Andalusian cross; and TB, Thoroughbred. The sex of the sample population was uniform, with 10 geldings (G) and body condition score (BCS) assessed using the Australian 0–5 scale as outlined by Leighton-Hardman in 1980. Horses were either kept on a night stabling routine (stabled at night, turned out in the day) or a day stabling routine (stabled during the day, turned out overnight).

**Table 2 animals-13-01065-t002:** Ethogram for stabled horse behaviour.

Behaviour Category	Behaviour	Definition
Eat	Eating/foraging	Seeking, nutritive chewing, and ingestion of substrate provided by owner
	Drinking	Consumption of water provided by owner
Stand	Standing positive (in/out/sideways/companion)	Horse standing with head and neck position higher than the wither (facing away from the stable door/facing towards the stable door, facing parallel to the stable door/facing a conspecific)
	Standing neutral (in/out/sideways/companion)	Horse standing with head and neck position approximately level with the withers (facing away from the stable door/facing towards the stable door, facing parallel to the stable door/facing a conspecific)
	Standing negative (in/out/sideways/companion)	Horse standing with head and neck position lower than the neck and withers (facing away from the stable door/facing towards the stable door, facing parallel to the stable door/facing a conspecific)
	Hind leg rested	Hind leg transferred into a non-weight-bearing position with slight flexion and incomplete contact of the hoof to stable floor
Lie	Lateral recumbency	The horse lying on one lateral side of its entire body, usually including head and neck. The head can briefly be held up by lateral flexion of the neck. Limbs can be held in different positions
	Sternal recumbency (in/out/sideways/companion)	The horse lying on the lateral side of the front and hindlimb closest to the ground and the ventrolateral part of the torso. The legs are more or less flexed and the head can be held upright or resting on the ground or its front legs
Locomotory	Step(s) forward	Step(s) transferring horse’s body in a forward direction
	Step(s) backward	Step(s) reversing horse’s body in a backwards direction
	Lateral step(s)	Step(s) transferring horse in a sideways direction that is neither forwards nor backwards
	Circling (partial/complete)	A full or partial movement of the horse’s body around the perimeter of the stable
Non-locomotory	Tail movement	Gentle lateral or vertical movement of the tail
	Tail swishing	Forceful lateral or vertical movement of the tail
	Non-locomotory leg movement	Movement of the legs that does not transfer the body forwards, backwards, or laterally
	Head raise	Raising of the head within the parameters of the horse’s stance (positive, neutral, or negative)
	Head lower	Lowering of the head within the parameters of the horse’s stance (positive, neutral, or negative)
	Head nod	A brief movement of the head in successive upward and downward motion
	Head tilt	A brief movement of the head on the lateral plane
	Lateral head movement	Movement of the horse’s head to either side
	Head shake	Rapid, rhythmic rotation of the head and neck along the long axis while standing with feet planted
	Scratching	Nibbling, biting, licking, or rubbing a part of the body
	Stretching	Rigid extension of the limbs and arching of the neck and back
Elimination	Urination	Forelegs slightly extended forward and hind legs extended backward and slightly spread, expelling of urine through the urethra. The penis is typically partially or fully relaxed from the prepuce
	Defecation	With tail raised, expelling of faecal matter through the anus
Other	Non-nutritive chewing	Chewing motion displayed by the horse without the consumption of food
	Yawning	A deep, long inhalation with mouth widely open and jaws either directly opposed or moving from side to side
	Human directed	Horse facing and/or approaching human on the outside of the stable
	Rolling	Dropping from standing to sternal recumbency, and then rotating one or more times from sternal to dorsal recumbency, tucking the legs against the body
	Tactile nose rubbing	Repeated movement of the horse’s nostrils against the bars of the stable wall on the lateral plane
	Conspecific aggression	Aggressive behaviour aimed at conspecific in neighbouring stall
	Pawing	A foreleg is lifted off the ground slightly, extended quickly in the forward direction, and followed by a backward, toe-dragging movement as if digging. This movement is typically repeated several times in succession
	Voluntary approach	Horse willingly approaches human on entry into stall
Unspecified	Ears back	Ears pressed caudally against the head and neck
	Ears forward	Ears held stiffly upright and forwards
	Ears neutral	Ears relaxed and facing laterally
	Ears 1F1B	Ears alternate forwards and backwards, remaining in this position for varied lengths of time
	Blink (eye)	Complete closure and reopening of the horse’s eyelid
	Half-close (eye)	Half closure and reopening of the horse’s eyelid
	Eyes closed	Complete and sustained closure of the horse’s eyelid
	No visual contact	Horse cannot be adequately seen on any camera

Ethogram of the 52 behaviours observed during the study with relevant behavioural categorisation. Unspecified behaviours were not used in the construction of time budgets as they can occur in conjunction with any of the other performed behaviours.

## Data Availability

Data are available upon reasonable request.

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
