# Peer review of "The Effect of Stabling Routines on Potential Behavioural Indicators of Affective State in Horses and Their Use in Assessing Quality of Life"

_animals, 2023, doi:10.3390/ani13061065_

Round 1
Reviewer 1 Report
Equine welfare is a very important topic and stabling is certainly a factor to consider. There is clearly a lot of useful data that was obtained in this study. Unfortunately, as currently presented, I fear the reader will be confused and struggle with a take home message. Ultimately, it seemed the data could not be used to determine if overnight stabling was detrimental to welfare or not. If this cannot be answered then it would be beneficial to go more in depth on the behaviors that might be predictive such as non-locomotive leg movement, nonnutritive chewing and yawning? Specific comments as follows:
Line 13-14: Awkward wording (reliable twice in same sentence)
Line 19: Should define affective state early on
Line 65: This is currently correctly termed asthma
Line 68-69: Reword. Recommend “Many facilities subdivide the land…”
Line 79-81: Reword this sentence; recommend also briefly defining this model
Introduction: The Introduction needs to better inform the reader why the behaviors that were used in this study were chosen. The intro should justify the specific outcome measures that were used and demonstrate why they are good indicators of horse welfare/ QOL. As written, the Intro does not do this.
Line 113: Approximately how deep were the boxes bedded?
Horse Housing: Need to include size of turn-out facilities/ range of sizes
Line 176-178: This concept should be introduced sooner- in the introduction.
Line 185-186: “with against” (confusing wording)
Line 188: The initial description says that horses will be compared stabled at night versus stabled during the day but here it also mentions comparing to horses stabled all of the time. Please clarify this.
Line 189: Please define the 7 groups
Line 235: Clarify if they performed more behaviors (because they were observed over more time) or if they performed more behaviors per time block
Line 258 and 262: I find the cluster dendograms very confusing/ not helpful in understanding the data. Strongly recommend a different type of chart.
Lines 310-329: Again the behaviors needs to be better described with information provided up front if these are negative or positive observations. Further it is confusing if the authors are trying to argue that horses stabled at night have a better QOL because they face away from the door with the head down (letting guard down- is this resting?) but also state that they may be more comfortable in a hard setting. This seems conflicting. Specifically, line 317-319- they are more sternal in overnight stabling and then unclear the point the authors are trying to make about giving this option in shared turnout?
Line 390-391: Since yawning and non-nutritive chewing have been shown to differ between stabled and non-stabled horses, it seems imperative that these be explored in this study and or warrant a paragraph in the discussion.
Discussion: Recommend more discussion surrounding night vs day in terms of light and activity. These factors seemed somewhat under analyzed throughout.
Author Response
Reviewer Comment |
Author Response |
Equine welfare is a very important topic and stabling is certainly a factor to consider. There is clearly a lot of useful data that was obtained in this study. Unfortunately, as currently presented, I fear the reader will be confused and struggle with a take home message. Ultimately, it seemed the data could not be used to determine if overnight stabling was detrimental to welfare or not. If this cannot be answered then it would be beneficial to go more in depth on the behaviours that might be predictive such as non-locomotive leg movement, non-nutritive chewing and yawning? Specific comments as follows: |
Thank you for this comment. Changes have been made to the introduction to improve the clarity of the message that is being communicated – in particular regarding potential behaviours that may be useful in the identification of affective state.
Like many common husbandry practices, stabling is unlikely to ever be considered “ideal” horse management, and it is about looking for opportunities of best fit for the individual rather than stating categorically one routine is better than the other.
We have increased the level of depth and discussion on the behaviours identified in your comment as these are the focus of the paper. |
Line 13-14: Awkward wording (reliable twice in same sentence) |
Thank you, the additional ‘reliable’ has been removed on line 14. |
Line 19: Should define affective state early on |
Definition of affective state included in line 19. |
Line 65: This is currently correctly termed asthma |
Thank you, terminology has been updated to reflect the change in line 66. |
Line 68-69: Reword. Recommend “Many facilities subdivide the land…” |
Wording has been updated in line 69. |
Line 79-81: Reword this sentence; recommend also briefly defining this model |
Rewording and definition have been included in lines 85-92. |
Introduction: The Introduction needs to better inform the reader why the behaviours that were used in this study were chosen. The intro should justify the specific outcome measures that were used and demonstrate why they are good indicators of horse welfare/ QOL. As written, the Intro does not do this. |
Thank you for this comment. However, the title and purpose of this paper is to try and identify which behaviours may be used to identify affective state, therefore we did not choose behaviours for this study, but tried to ascertain if there were any behaviours of significance when horses were housed in different ways and whether these behaviours may be useful in identifying affective state with further research and validation. |
Line 113: Approximately how deep were the boxes bedded? |
Text has been included in line 142 to state bedding depth. |
Horse Housing: Need to include size of turn-out facilities/ range of sizes |
Content updated on lines 137 and 140 to include additional information regarding size of housing facilities. |
Line 176-178: This concept should be introduced sooner- in the introduction. |
Concept has now been included in the introduction on lines 109-113. |
Line 185-186: “with against” (confusing wording) |
Thank you – “against” deleted from line 227. |
Line 188: The initial description says that horses will be compared stabled at night versus stabled during the day but here it also mentions comparing to horses stabled all of the time. Please clarify this. |
Wording updated to improve clarity – time budget of full-time stabled horses taken from Kiley-Worthington (1989), not from additional research done in this project. |
Line 189: Please define the 7 groups |
Groups included on line 231-232. |
Line 235: Clarify if they performed more behaviours (because they were observed over more time) or if they performed more behaviours per time block |
Updated on lines 281-282. |
Line 258 and 262: I find the cluster dendrograms very confusing/ not helpful in understanding the data. Strongly recommend a different type of chart. |
Thank you for your comment. On further discussion the authors have decided that the use of the dendrogram is something we would like to retain, given its further explanation and use in the discussion on lines 353-358. However, we have made its purpose for inclusion clearer by including additional information in lines 396-403 and lines 443-450. |
Lines 310-329: Again, the behaviours need to be better described with information provided up front if these are negative or positive observations. Further it is confusing if the authors are trying to argue that horses stabled at night have a better QOL because they face away from the door with the head down (letting guard down- is this resting?) but also state that they may be more comfortable in a hard setting. This seems conflicting. Specifically, line 317-319- they are more sternal in overnight stabling and then unclear the point the authors are trying to make about giving this option in shared turnout? |
Sternal recumbency – stated on line 367
Locomotory leg movements – added into line 395-396.
Frequency of ear movement – expanded upon in lines 414-416
Non-nutritive chewing – added to line 442-443.
Yawning – added to line 460.
Comments on sternal recumbency updated on lines 368-370 to improve clarity. |
Line 390-391: Since yawning and non-nutritive chewing have been shown to differ between stabled and non-stabled horses, it seems imperative that these be explored in this study and or warrant a paragraph in the discussion. |
Thank you for this comment. Additional information regarding non-nutritive chewing has been included in lines 433-450. |
Discussion: Recommend more discussion surrounding night vs day in terms of light and activity. These factors seemed somewhat under analysed throughout. |
Additional information regarding lighting included in lines 163-168.
Activity in the stable in terms of behaviours performed is outlined in both the ethogram and the discussion. |
Reviewer 2 Report
Thank you for an interesting paper that highlights important welfare aspects on stabled horses and introduces a new research question that contribute to the area of horse management. The aim was to investigate the effect of stabling on equine behavior by observing night stabled and day stabled horses and by comparing their behavioral repertoire. It is concluded that non-locomotory leg movement, and ear position and frequency of ear movement may be used as potential welfare indicators and measures of equine Quality of Life.
The paper is well-written but each section require improvements. I miss an introduction that points more directly to the aim; clearly described hypotheses; an adequate description of the overall study design; statistical models that answer to stated hypotheses; and a discussion on the limitations of the study. I therefore recommend major revisions of the paper. I will provide further comments in the attached file and I hope that this feedback will contribute to a final version of this paper.

Author Response
Reviewer Comment |
Author Response: |
The importance of improving equine welfare is well covered in the introduction, as are how stabling may affect welfare negatively. However, I think that a brief summary on stabled behaviours and differences between behaviours during day and night should be included. |
Thank you, this has been expanded upon in lines 73-78. |
Line 66-67: Very similar to line 60-61 – lack of understanding may impair welfare… |
Thank you, lines 60-61 updated. |
Line 68-69: Please include a reference for this statement. |
Amended and reference included. |
Line 82-83: Please explain the concept of Quality of Life and already included behaviours since the paper aim to identify behavioural changes that may be suitable QoL measures. |
Updated information in lines 96-102 discussing that there is no standardised EQoL framework in place yet, hence the need for increased research into behaviours that may contribute to this. |
I also miss hypotheses that are tested with the different statistical methods. Please rewrite the aim so it only includes what is tested in this paper and add hypotheses. |
With respect, hypotheses tend to be implicit when describing statistical testing and reporting statistical results. The authors have therefore not included explicit hypotheses in this paper, similar to other recently published journal papers in this area (e.g., Merkies, K., Crouchman, E., & Belliveau, H. (2022). Human ability to determine affective states in domestic horse whinnies. Anthrozoös, 35(3), 483-494. |
There is no overall description of the study design which I think is required |
This is stated in line 176. Further information regarding the study design can be found in lines 136-144. |
Was a pilot study performed first (line 141-143)? If so, were that data used? |
Thank you for this comment, information added to clarify this point in lines 182-184. |
Please state how the horses were recruited for this study, how they were determined healthy and how they were assigned a stabling routine. |
Additional information added in lines 117-119. |
Were the horses moved to the stabling facility from the owners? If yes, how were they acclimatized and for how long? Or was the Charles Sturt University Equine Boarding Centre their home environment? If yes, state this (line 138 is not clear enough). |
Additional information added in lines 117-119, 124-125 and 147-152. |
What were their previous stabling routines were (day/night, bedding, knowledge of the other horses in the herd)? |
Additional information added in lines 147-152. |
Please include information on how long in total each horse was kept in its assigned stabling routine, since only information on how long they were recorded is included. |
Additional information added in lines 147-152. |
Please include information on lightning routines. Was it only day light or lamps during the day? When was the infrared LEDs turned on? When was it dark? |
Information added in lines 163-168 regarding lighting routines. |
Line 135: The information about Rcmdr vR3.6.2 belongs to section 2.7. |
Thank you for this comment, the information has been moved to section 2.7 |
Line 141-142: How was the number of cameras decided? |
Already stated in lines 180-182. |
Line 145: Repetition |
Thank you – repetitive information has been removed from lines 187-188. |
Line 177: Which ethogram is used? In line 185 it is referred to Table 2 which is an ethogram – is that the same? |
Yes, that is correct, which is why Table 2 is referred to at the conclusion of the sentence on line 220. |
Line 188: Which are the horses being stabled full time? Ref 22? Suggest rewriting the sentence |
Thank you – sentence re-written. |
Line 190-193: Which behaviours were excluded and not considered as performed behaviour? |
Thank you for this comment, information on lines 230-237 has been updated to further clarify this information. |
Line 194-195: The information about Rcmdr vR3.6.2 belongs to section 2.7. |
Content has been removed from section 2.6. |
In section 2.7 normality of the dataset is assumed based on reference 23. However, that reference discusses that data transformations for normality assumptions are not required when having large datasets and performing linear regression statistics. Have any normality testing such as the Shapiro Wilks test or evaluation of histograms/QQ-plots been made? |
Thank you for bringing my attention to this. When consulting with the statistician regarding the data, he said the normality of the data had been assumed, however on going back through the different methods of statistical analysis it was found that a Shapiro Wilks test had been conducted and data were found to be parametric. Line 248-249 have been updated to reflect this. |
How many t-tests are performed? I think you should consider including correcting for multiple t-tests or state why it is not needed. |
Only one unpaired t-test was performed per behaviour (total behaviours day vs night), therefore a correction was not deemed necessary. |
Have you considered other statistical models such as ANOVA to evaluate the stabling effect and where the random effect of ‘horse’ can be included? |
An ANOVA was not conducted due to the small number of horses observed, and that the analysis was done on results of total night stabled population vs. day stabled population, rather than individual totals of each horse. |
Rcmdr is a graphical user interface and I guess R is the program you have used. Please include this information and refer correctly to R. Please also include names of R packages used to perform all computations. |
Thank you, this has been amended. |
Please state that 14.8 is the mean and 1.1 is the standard deviation (line 216). Also include mean (SD) behaviours per group (day vs night) in line 217. |
Information has been updated in lines 261-263. |
I think Figure 1 should be in colour or with patterns for the categories since it is difficult to differ between the grayscales. |
Thank you for this comment. It has been noted and the graph has been updated accordingly to assist with differentiation between the categories. |
Please describe the main findings of the hierarchical clustering analysis for the reader. If not used to HCA the plots may be difficult to interpret. |
Thank you, additional information regarding the dendrograms in lines 395-403 and lines 442-450 have been included to discuss relevant behavioural clusters. |
The beginning of the discussion I think belong to the introduction (line 269-281). For increased readability, the discussion may begin with stating the main findings of the paper and then proceed to discuss these findings specifically in relation to previous research. |
Lines 316-329 replaced. |
I also think the discussion could benefit from including how the behaviours identified in this paper can aid in assessing Quality of Life since it is a part of the aim and title |
Thank you, this point has been expanded upon in the discussion in relation to behaviours identified on lines 410-496. |
Please include a discussion of limitations of this study. Could you have designed the study in another way and for instance included more horses? Is there inter-individual variation in behaviours that may affect the results? What if you sampled more than 10 minutes per block? What about the effect of observer presence? |
Limitations regarding inability to use SBR outlined on lines 346-348.
Further discussion on study limitations included on lines 497-504.
Recordings were done via cameras installed prior to horse entry into stable; observers were not on site for recordings therefore there was no observer presence when stabled. |
Please note that the DOI-links given for most of the references contain “-org.ezproxy.csu.edu.au” making the links unavailable if you are not a member of Charles Sturt University. It is recommended to change to original DOI. |
Thank you for pointing this out, this has been rectified. |
Are there any owner consent from the horse owners? |
Yes, owner consent forms have been provided to Animals, and text has been included to reflect this on line 113. |
Usually, the data used for statistical computations – in this case the frequency tables in Excel, are made publicly available or available from the authors upon reasonable request. Please elaborate on your statement “not applicable”. |
Thank you for this point, we have updated the section so that data is available from the authors upon reasonable request. |
Round 2
Reviewer 1 Report
The authors edits have greatly improved the quality of the manuscript and it will now be easier for authors to understand the premise of the study and the meaning of the various observations.
Line 471-474: Recommend rewording this sentence to remove 'this is evidenced by' and simply state that one horse transitioned to full time stabling and thus the horses data had to be excluded.
Author Response
Reviewer Comment |
Author Response |
The authors edits have greatly improved the quality of the manuscript and it will now be easier for authors to understand the premise of the study and the meaning of the various observations. |
Thank you very much for your supportive words. |
Line 471-474: Recommend rewording this sentence to remove 'this is evidenced by' and simply state that one horse transitioned to full time stabling and thus the horses data had to be excluded. |
Thank you, this has been amended on line 485. |
Reviewer 2 Report
Thank you for providing a revised version and a cover letter. Please see further comments in the attached document.

Author Response
Reviewer Comment |
Author Response: |
The introduction is improved and well written. However, the study compares day and night behaviours, so more information on behaviours during day vs night is required, as are the influence of the circadian rhythmicity and presence of light on behavior. Or are there no such studies yet – if so, this needs further clarification (except line 76)? |
Additional information regarding light and circadian rhythm added to line 77-79. |
I do not agree with the authors on the choice to exclude hypotheses. The statistical methods included in this paper are of two types; describing statistics where no hypotheses are needed (done with HCA) and hypothesis testing (done with unpaired t-tests). Therefore, hypotheses should be included to inform the reader what you are testing with the t-tests. Your aim is to determine the effect of stabling routines on equine behaviour, and your hypotheses would for instance be that certain behaviors are performed significantly more during night than day. |
Thank you for your thoughtful comment on this matter. A null hypothesis has been included on line 108-109. |
Line 117: Please add mean and standard deviation to explain the numbers 11.8 ± 4.4 years, suggestion: “mean (standard deviation, SD) years of 11.8 (4.4)” |
Amendments made as per your suggestion on line 121. |
Line 120: Suggestion: “The mean (SD) Body Condition Score (BCS) was 3.35 (0.45)”, since you now explained mean and SD in line 117. This is suggested to be applied to all other lines where mean (SD) are described. |
Thank you for this suggestion. It has been taken on board and amendment made in line 124-125 and other areas referencing mean (SD). |
Thank you for clarifying details in section 2.2, 2.4, 2.5 and 2.6. |
Thank you for your comments and time reviewing this paper. |
Line 228-229: As already commented on in previous report, R should be referred to correctly. It is not a package but a program. Suggest to move this sentence to section 2.7. |
Thank you for this correction. Wording has been changed from package to program and moved to section 2.7. |
Section 2.7 does not contain a correct citation for R and the used R packages despite the authors responding that it does. What package did you use for HCA? |
The author apologises for the error here. Section 2.7 has been updated to include the correct citation for R and description of package used to generate the HCA. |
Regarding the series of unpaired t-tests, if I understand it correctly you perform one t-test for each behaviour and is that 52 tests in total? When performing multiple t-tests you increase the risk for Type I error, why a statistical correction (such as the Holm-Bonferroni or Tukey’s correction method) must be performed to handle the risk that some of the significant p-values may be significant by chance. If you perform 50 tests you risk that 2-3 of them are significant by chance, which may affect the whole outcome of the paper. |
Thank you for your further comments on this section. |
In Figure 1 I still struggle to see the differences? Is it correct that behaviours belonging to the locomotory category are not present in the figure? I only see 5 blocks for day stable regime and 6 blocks for night stable regime. |
Due to the elimination and lying sections having fewer/less frequent behaviours, these sections are smaller. The chart has been adjusted to separate out the categories for improved clarity, given the patterns were found to be insufficient. |
Since the figure is not visually clear I suggest to include a table with means and confidence intervals for each behaviour category (or even each behaviour) to support Figure 1 and 2. If not in the paper, then as a supplementary file. |
|
Line 281: “showed increased significance in frequency” should be changed to “showed significant increases in frequency” |
Amendment made on line 295. |
Line 321: You have not explained SBR? |
Thank you for this correction, previous content regarding Spontaneous Blink Rate was removed from the Discussion section resulting in the incorrect use of the acronym left behind. |
Line 474-477: One of your biggest limitations is the small sample size and I think that should be included in the limitation section. |
Thank you for this suggestion, this limitation has been included on lines 490-491. |
Round 3
Reviewer 2 Report
Thank you for the revised paper and the clarifying responses. The revisions have improved the paper further and I only have some minor comments.
Lines 107-108: Thank you for stating the null hypothesis, but when written as "it was hypothesized" the reader may not understand it is the null hypothesis. Please rewrite or clarify, since your significant p-values confirm you hypothesis.
Line 120: It is the first time you mention SD, please include standard deviation as well.
Lines 238-246: Thank you for clarifying this section.
Author Response
Thank you for the revised paper and the clarifying responses. The revisions have improved the paper further and I only have some minor comments.
Lines 107-108: Thank you for stating the null hypothesis, but when written as "it was hypothesized" the reader may not understand it is the null hypothesis. Please rewrite or clarify, since your significant p-values confirm you hypothesis.
Thank you, lines 107-108 have been amended to improve clarity.
Line 120: It is the first time you mention SD, please include standard deviation as well.
Thank you, this has been amended on line 120.
Lines 238-246: Thank you for clarifying this section.
Thank you for your time and effort in reviewing this submission, it is greatly appreciated.